# The Temporal–Spatial Dynamic Distributions of Soil Water and Salt under Deep Vertical Rotary Tillage on Coastal Saline Soil

**Wenxiu Li** [1,2], **Jingsong Yang** [1,*], **Chong Tang** [1], **Xiaoyuan Liu** [1], **Wenping Xie** [1,*], **Rongjiang Yao** [1] and **Xiangping Wang** [1]

1    State Key Laboratory of Soil and Sustainable Agriculture, Institute of Soil Science,
Chinese Academy of Sciences, Nanjing 210008, China

2    University of Chinese Academy of Sciences, Beijing 100049, China

*    Correspondence: jsyang@issas.ac.cn (J.Y.); wpxie@issas.ac.cn (W.X.)

**Abstract:** Different from the traditional deep tillage, deep vertical rotary tillage can smash deep soil without disturbing the soil layer, which improves soil water infiltration and promotes salt leaching. This has gradually been applied in the research into saline alkali improvements. However, there is limited knowledge about the effect of deep vertical rotary tillage on the temporal–spatial distributions of soil water and salt under the shallow underground water level. Therefore, a preliminary soil column experiment was carried out to explore the results of water and salt movement under three different tillage methods: traditional rotary tillage depth of 20 cm (XG−20), deep vertical rotary tillage depth of 20 cm (FL−20) and deep vertical rotary tillage depth of 40 cm (FL−40). The temporal–spatial variation in soil water and salt was analyzed. The results showed that the average infiltration rate of FL−40 increased by 1.25 and 0.46 cm h$^{-1}$ in 0−40 cm soil layer compared with that of XG−20 and FL−20. At the same time, soil water content was also increased, and the order of soil water content was FL−40 > FL−20 > XG−20. With the increase in tillage depth, the desalinization rate and the soil electric conductivity (EC) of FL−40 were increased and decreased, respectively. The FL−40 treatment's average desalinization rates increased by 16.32% and 13.99% compared with XG−20 and FL−20 treatments' in 0−60 cm soil layer. In conclusion, FL−40 had a better effect on regulating water and salt. The deep vertical rotary tillage provided an effective method for the control and optimization of water and salt in coastal saline soil.

**Keywords:** deep vertical rotary tillage; temporal−spatial dynamic distributions of soil water and salt; coastal saline soil





## 1. Introduction

Soil salinization and secondary salinization in coastal area are serious issues due to the influence of topography and climate and improper field management. Reasonable water and salt regulation measures are important ways to improve the saline alkali barrier in this region [1]. The aim is to control the water and salt in the root zone of crops within a certain tolerance range, avoid water and salt stress, and increase crop yield.

Previous studies have shown that tillage methods can control the dynamics of water and salt by changing soil's physical structure to inhibit soil evaporation and improve the infiltration and leaching of salinity [2]. Therefore, tillage can be considered as one of the most important agricultural management measures to improve saline alkali land. However, many previous studies have shown that soil bulk density and soil water can be significantly affected by various tillage methods, thus affecting soil surface water migration [3–7]. Cai et al. (2014) [8] found that the average tillage depth was 16.5 cm in China. A shallow and dense tillage layer deteriorates soil's physical and chemical properties such as aeration, water permeability, water storage and fertilizer storage, which can restrict the growth and development of crop roots, creating a limiting factor for crop yield and harvest [9]. As

can be seen from the above studies, the plough layer formed by traditional tillage reduces water infiltration and hinders salt leaching. The dense soil layer reduces soil water storage capacity and increases water and salt stress. Therefore, it is crucial to further optimize tillage measures to improve the saline alkali barrier and water-use efficiency in coastal saline soil.

In recent years, the deep vertical rotary tillage technology developed has improved this situation. This is different from the two common deep tillage methods. One is deep ploughing, which pushes fertile soil into the lower layer, reducing soil fertility [10]. The other is deep pine, which has a limited area of loose soil. Soil block is larger after tillage, affecting crop growth [11]. However, deep vertical rotary tillage overcomes these problems. The machine includes six vertical spiral drills, which can vertically smash up to a maximum soil depth of 1 m without disturbing the soil layer [12]. After tillage, the soil clods were crushed to a higher degree, forming a thick loose layer. Therefore, without the hindrance of the topsoil, the soil water permeability increases, and the topsoil salt is more likely to be washed into the deep soil. The upward movement of salinity in the lower layers is reduced by the cutting of capillaries [13]. Previous studies have shown that deeptillage increases soil porosity, water conductivity and infiltration rate by reducing soil bulk density and penetration resistance, leading to a higher water content in deep loosening layers and reducing soil salinity [14–18]. Therefore, deep vertical rotary tillage has potential advantages in the improvement and utilization of mildly saline alkali land. Although there have been a lot of studies on the water–salt rule under deep vertical rotary tillage, they were all concentrated in the arid area of Northwest China, where the groundwater level is deep and the distribution of soil water and salt is affected by rainfall and surface irrigation. However, the factors affecting the distribution of soil water and salt in coastal areas are complex. The groundwater level in this area is shallow and highly mineralized, and groundwater and soil water frequently interact. It is very important to clarify the dynamics of water and salt at a shallow groundwater level for the further optimization regulation of water and salt. Under shallow-groundwater-level conditions, the dynamic distribution of water and salt under deep vertical rotary tillage has rarely been studied.

This study's ultimate aim was to evaluate the effect of deep vertical rotary tillage on water and salinity regulation in saline alkali soil with a shallow underground water level. In this paper, it was hypothesized that deep vertical rotary tillage could improve soil water infiltration performance, increase soil water storage capacity and increase salt leaching efficiency. The above effects were more significant with the increase in tillage depth. Therefore, a soil column experiment was carried out to study the characteristics of the temporal–spatial dynamic distributions of water and salinity under different tillage methods on coastal saline soil. This provided a basis for the study and regulation of water and salinity movement under deep vertical rotary tillage on coastal saline soil.

## 2. Materials and Methods

### 2.1. Study Area, Sampling Sites and Sample Analysis

Saline soil was sampled from a depth of 0−30 cm at the Dong Tai Ecological Research Station of Coastal Wetland (Figure 1), Chinese Academy of Sciences, located in Yan Cheng, Jiangsu Province, China. This area is a subtropical monsoon maritime climate, with distinct seasons and rain and heat over the same period. The annual average air temperature and rainfall are 15.0 °C and 1061.2 mm, raining enough, with 220 frost−free days and 2130.5 h sunshine time. The region is severely affected by soil salinization.

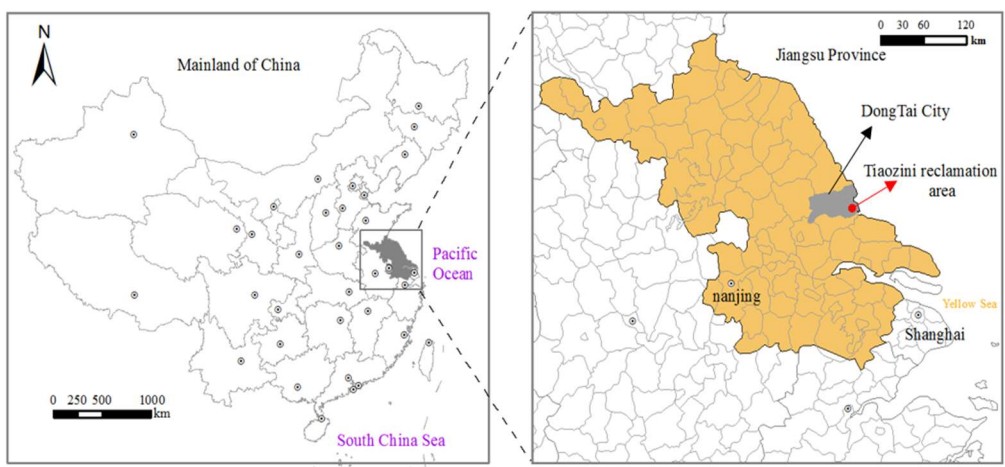

**Figure 1.** Study area location.

Soil samples were air-dried at room temperature (25 °C) and then sieved using a 2 mm sieve to remove coarse debris and stones. The initial analysis indicated a medium soil salinity; soil water content was determined by drying method; soil pH and soil electrical conductivity (EC) were measured in a supernatant of 1:5 (*w/v*). Soil-water mixtures were measured using a pH meter and an electronic conductivity meter (Mettler Toledo, OH, USA). The total salinity content was measured as residue drying (Table 1). The desalination rate was calculated as the percentage difference between salinity values measured before and after the experiments.

**Table 1.** The physical and chemical properties of soil were tested.

| pH | EC ($\mu s\ cm^{-1}$) | Salinity Content ($g\ kg^{-1}$) | Sand Content (%) | Silt Content (%) | Clay Content (%) | Soil Texture |
|---|---|---|---|---|---|---|
| 7.98 | 1600 | 3.35 | 13.2 | 61.5 | 25.3 | Silt loam |

Note: EC, soil electrical conductivity of soil extract with a soil–water ratio of 1:5.

### 2.2. Soil Column Experiment

Soil columns were prepared using transparent polymethyl methacrylate (PMMA) cylinders (15 cm diameter, 130 cm height). Prior to filling the column, a 5 cm thickness of fine quartz sand was placed at the bottom. To obtain homogenous soil bulk density ($1.35\ g\ cm^{-3}$), the sieved dry soil was poured into each column in 10 cm sections; the soil surface of each packed layer was stirred before adding the next to prevent stratification. The soil was packed in each column until a height of 110 cm was achieved. Soil columns were irrigated according to the water content suitable for cultivation, and soil was cultivated after irrigation. After tillage, the soil columns were placed under infrared and the soil was allowed to dry so that wettability peaks could be observed. A tape measure (length, 110 cm) was vertically fixed to the outer wall of each soil column; this was used to record the location of the wetting front during infiltration. The sampling holes were distributed at 10, 20, 40, 60, and 80 cm vertically downward from the soil surface. Four groups of the sampling holes were set on the bottle body. Three treatments were established, and each treatment was repeated three times: traditional rotary tillage 20 cm (XG−20), deep vertical rotary tillage 20 cm (FL−20) and 40 cm (FL−40). The schematic diagram of experimental treatment is shown in Figure 2. To simulate the field cultivation as closely as possible, a small rotary cultivator (helix blade diameter of 10 cm) was used to simulate field rotary tillage, and a hand electric drill with auger (helix blade diameter of 4 cm; the machine´s speed was the same as that of the field deep vertical rotary machine: $460\ r\ min^{-1}$) was used to simulate deep vertical rotary tillage. To avoid marginal effects, tillage was carried out in the center of the soil column without touching the wall.

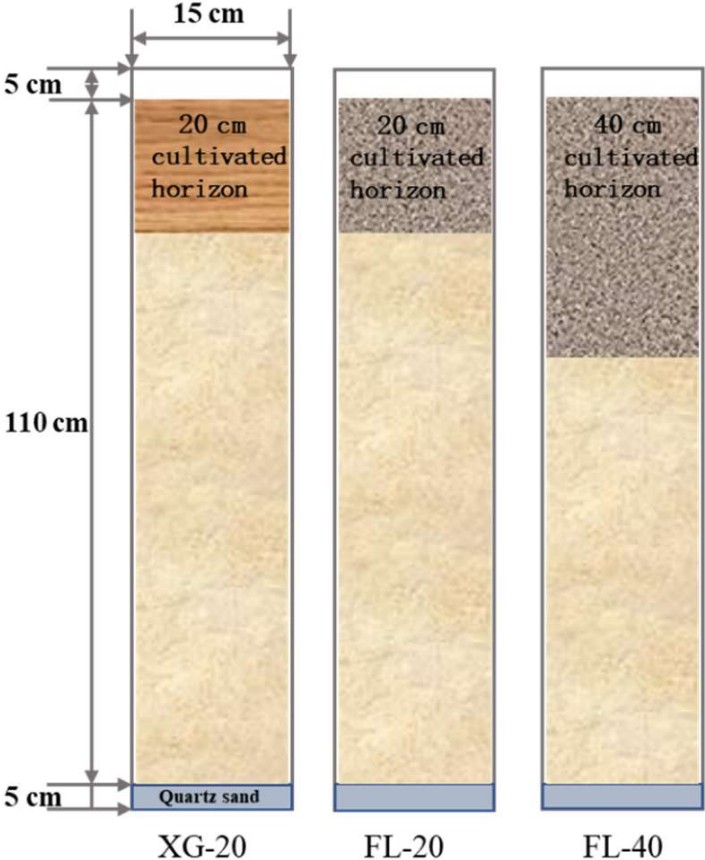

**Figure 2.** Schematic diagram of soil column experiment (traditional rotary tillage 20 cm (XG−20), deep vertical rotary tillage 20 cm (FL−20) and 40 cm (FL−40)).

The infiltration experiment was carried out at the time of the first irrigation. The wetting peak was observed every five minutes at the beginning of the infiltration, then every half hour, and every hour after the infiltration stabilized. A 37−day simulation test was conducted after the end of the infiltration experiment, during which two irrigations were carried out. The average soil water content in 0–60 cm soil layer of the three treatments before irrigation was 0.285 g g$^{-1}$. The time of the first irrigation was selected as the first day of the simulation, and the second irrigation was carried out on the 21st day. The meteorological data of the meteorological station in the test site from 2000 to 2012 were used as reference. The average monthly precipitation of 140 mm in the maize growing season from April to August was used as the irrigation amount for the experiment. The irrigation amount was 1.24 L for both times, and variable pressure head irrigation was adopted. On the 0, 4, 19 and 37 days, samples were taken from 10, 20, 40 and 60 cm soil layers, respectively. Approximately 10 g of soil was removed from the sampling hole and then backfilled with moist mother soil. Meanwhile, water samples from Mariotte's bottles were taken to determine soil water content and electric conductivity.

Mariotte's bottle was used at the 110 cm layer to measure the underground water depth. After irrigation, the infrared lights were placed 20 cm from the soil surface to simulate evaporation. The irradiation lasted for eight hours each day from 10:00 a.m. to 18:00 p.m. Evaporation was weighed at 10 a.m. every day, and the difference between the two weights was the daily evaporation (Figure 3).

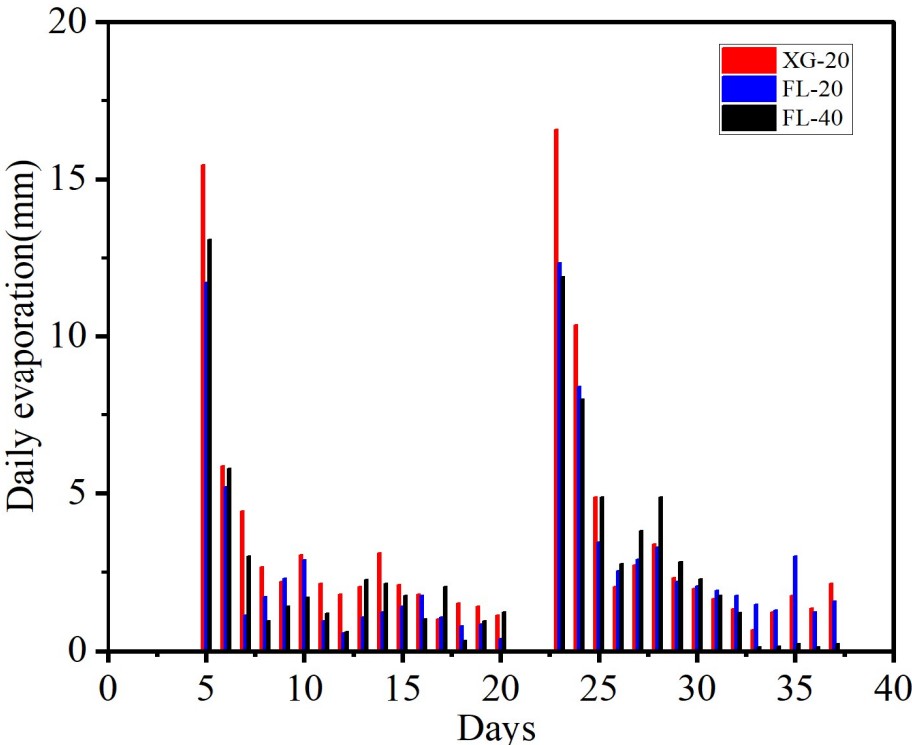

**Figure 3.** Daily evaporation under different tillage treatment methods.

### 3. Statistical Analysis

Microsoft Excel 2019 was used to calculate the means and standard deviations. SPSS was used to analyze the mean values of the data by one-way ANOVA (Analysis of Variance) with Student−Newman−Keuls multiple-comparison test at the level of $p < 0.05$. Before the ANOVA, the data were tested for equality of variances by Levene's test. The figures were drawn using Origin 2022b and Microsoft Excel 2019.

### 4. Results and Discussion

#### 4.1. Effect of Deep Vertical Rotary on Soil Water Infiltration

Soil water infiltration is an important process of soil water and salt distribution [19]. Figure 4 shows the dynamic changes in the wetting front with time under different treatments. It can be seen that different tillage methods have a significant influence on soil infiltration. By comparing the soil infiltration of 0−40 cm soil layers, it can be seen that the infiltration difference between different treatments gradually increases over time, and then gradually decreases until it becomes stable. The Table 2 shows the average infiltration rate of each soil layer under different treatments. It can be concluded that the infiltration rate of the XG−20 treatment was significantly lower than that of the other two treatments in 0−20 cm soil layer. In 20−40 cm soil layer, the infiltration rate of FL-40 treatment was significantly higher than that of FL−20 and XG−20. Although the tillage depth of FL-20 was the same as XG−20, the infiltration rate of FL−20 was still higher than XG−20. As the saturated water content of FL-20 was higher than that of XG−20, the higher water potential increased the infiltration rate into the underlying soil [19]. The average infiltration rate was FL−40 > FL−20 > XG−20 in 0−40 cm soil layers. The results showed that tillage depth was an important factor affecting soil infiltration, and the soil infiltration rate increased with the increase in tillage depth. When the vertical rotation depth is 40 cm, the soil infiltration performance can be significantly improved.

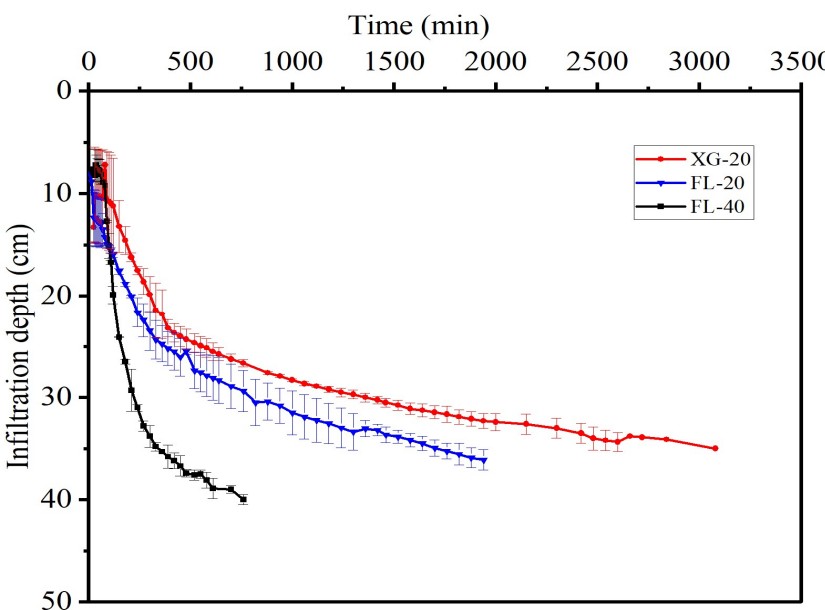

**Figure 4.** Wetting front under different treatments.

**Table 2.** The infiltration rate at different layers.

| Treatments | XG-20 (cm·h$^{-1}$) | FL-20 (cm·h$^{-1}$) | FL-40 (cm·h$^{-1}$) |
| --- | --- | --- | --- |
| 0–20 cm | 1.12 | 2.19 | 2.62 |
| 20–40 cm | 0.27 | 0.47 | 1.28 |

In this study, we found that deep vertical rotary tillage improved soil infiltration performance in the cultivated layer. This was consistent with the results of Jiang et al and Wei et al. [20,21]. The soil structures of saline–alkali land and the soil water infiltration capacity were poor. This was an important problem in the process of saline–alkali land improvement. Water infiltration performance played an important role in the leaching of soil salt. Deep vertical rotary tillage can loosen the soil, make the soil particles fine and even, and promote water infiltration. Compared with the treatment of different tillage depths, the water infiltration of the FL−40 treatment was the fastest. The results of this study are consistent with those of previous studies. Yang and Ma et al. proved, through both laboratory and field experiments, that the water infiltration rate of deep vertical rotary tillage was significantly higher than that of traditional rotary tillage. Other studies have found that when the depth of traditional cultivation reaches 20 cm, it cannot break through the plough plate formed by years of cultivation, which hinders soil water infiltration and salt leaching [15,22]. In this study, deep vertical rotary tillage was used to reduce soil bulk density and increase soil infiltration rate in topsoil. Soil water infiltration rate increased with increasing tillage depth.

*4.2. Temporal–Spatial Dynamic Distributions of Soil Water*

During the two irrigation and evaporation processes, there were differences in the temporal–spatial dynamic distribution of soil water on 0−60 cm layers in each treatment (Figure 5). With the occurrence of irrigation and evaporation, the water content of 20 cm soil in each treatment fluctuated greatly and showed significant differences. Compared with the water content on the 4th day after irrigation, the range of decrease in soil water content (19 and 37 days) in 20 cm were both FL−40 > FL−20 > XG−20, and the reductions were 13.74%, 8.36%, and 7.86% and 17.19%, 14.44%, and 2.81%, respectively. The reduction in soil water content in the FL−40 treatment was significant ($p < 0.05$) on the 37th day. The soil water content of FL−40 decreased faster due to evaporation. However, in the 40 and 80 cm soil layers, the water content of the FL−40 treatment was always higher than that

of other treatments. The average soil water content at 40−60 cm significantly increased ($p < 0.05$) with FL−40 treatment. The average water contents of FL-40, FL−20 and XG−20 in 40−80 cm soil layers were 0.428, 0.415 and 0.412 cm cm$^{-3}$, respectively.

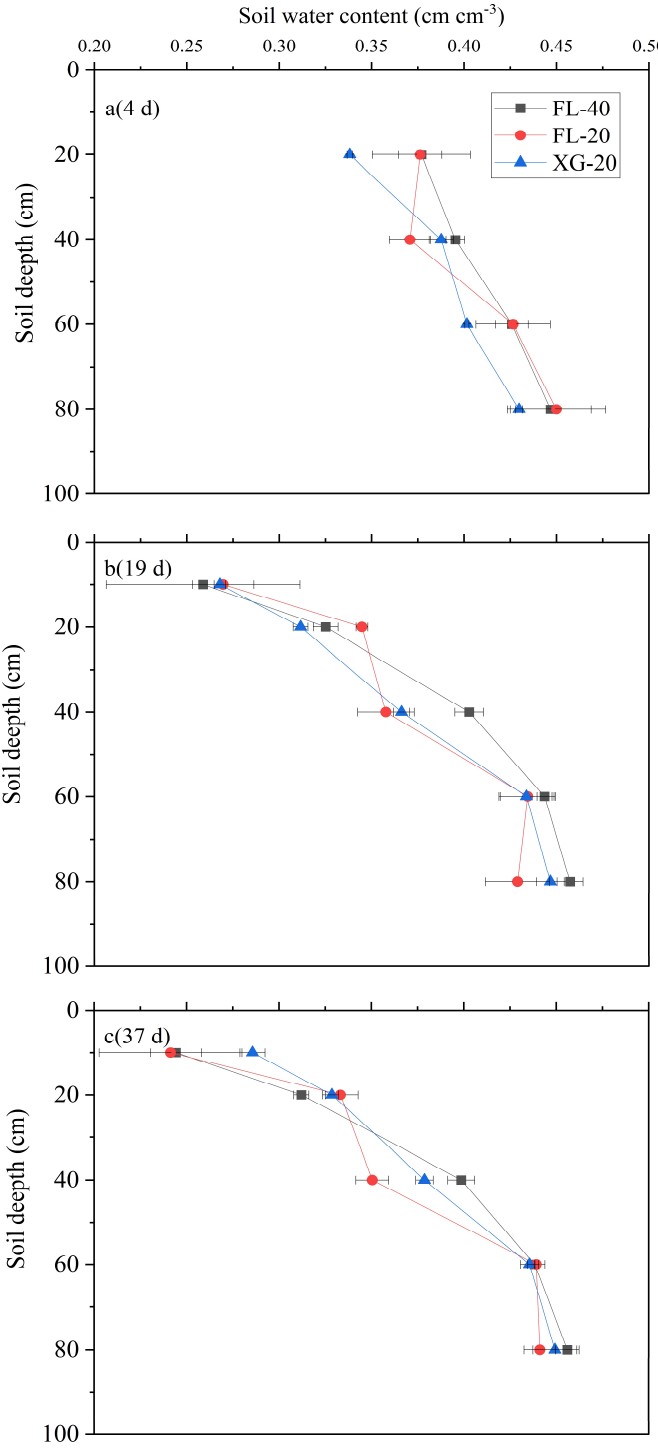

**Figure 5.** Dynamic distributions of soil water content in the soil profiles under different treatments.

In salinized soil improvement process species, soil water management is an important process to alleviate salt stress [23]. Soil water distribution was significantly affected by deep vertical rotary tillage. Our study found that the soil that was loose after deep vertical rotary tillage had better air permeability. When there was no land cover, the soil water content in the 0−10 cm surface layer, subject to high atmospheric evaporation force, rapidly decreased to form a dry soil layer, which was consistent with the study of Li et al. In the

20−40 cm soil layer, the water content of the FL−40 treatment was always higher than that of the other treatments. The deep vertical rotary tillage can crush the bulk soil to form a uniform and fine granular structure, increasing the porosity. When supplied with water, the water storage will be higher than other treatments [24–27]. In addition, the presence of a dry soil layer in the top layer could block the rise in capillary water in the lower layer, so the soil water content in the 40−60 cm layer was increased. Due to the separation of soil particles from the underlying soil structure, the capillary tube is interrupted, the rising of the capillary water to the surface soil is reduced, the evaporation loss of water is reduced, and the water in the deep soil is preserved at the same time [27].

The force between soil particles and water molecules determines the water-holding capacity of soil, which is affected by soil porosity, soil texture and other factors [18,28–31]. The soil was porous, with a reduced bulk density, increased porosity, and expanded soil water storage capacity, which was an important reason for the increase in soil water content in 40 and 60 cm deep layers. A high number of studies have shown that deep vertical rotary tillage can improve soil structure and increase soil water-stable aggregate content [30,32,33]. Wang et al. [34] observed, using scanning electron microscope, that the soil micromorphology after silty ridge cultivation had the characteristics of a smooth surface, larger specific surface area and richer pore distribution. Fine soil particles have a large pore surface area and adsorption capacity [35], which can improve soil water-holding capacity. This was also an important reason for the increase in soil water content. With a water supply, compared with the traditional rotary tillage, when the deep vertical rotary tillage layer is 40 cm, the deep and loose tillage layer was conducive to water infiltration; thus, more water is stored [23,31]. When continuous evaporation occurred, the deep soil moisture was less affected by evaporation, and the deep soil moisture was more conducive to preservation, thus increasing the deep soil moisture contents at 40 and 60 cm. The results were consistent with those of previous studies. A large number of studies showed that deep vertical rotary tillage could reduce soil bulk density and increase soil water content [20,31,36]. When drought persists in the field, the deep water stored in the deep vertical rotary tillage will play an important role in crop growth and development. However, traditional rotary tillage can further aggravate salt stress during droughts [37,38]. In conclusion, deep vertical rotary tillage can increase the water content in soil at 40 and 60 cm depths, and water storage can be effectively carried out during droughts.

*4.3. Temporal–Spatial Dynamic Distributions of Soil Electric Conductivity*

Throughout the whole experiment, the dynamic changes in EC under different treatments were significantly different (Figure 6). The EC of XG−20 greatly fluctuated in the 20 cm soil layer. On the first four days after the first irrigation, the ECs of XG−20, FL−20 and FL−40 treatments were significantly reduced by 13.8%, 25.1% and 36.7%, respectively. The reduction in FL−40 was significantly ($p < 0.05$) higher than that of the other two treatments. After 19 days, the EC of XG-20 increased by 6.4% compared to after four days. FL−20 and FL-40 treatments decreased by 6.0% and 7.1%. After the second irrigation, the EC of the 0−40 cm soil layer further decreased in each treatment. However, in the 60 cm soil layer, the salt from each treatment began to accumulate, and the cumulative degree was XG−20 > FL−20 > FL−40. This showed that the desalinization effect of FL−40 is better than the other two treatments when using the same irrigation amount. It may be that FL−40 treatment reduced bulk density and cut off the capillary, promoting salt leaching and inhibiting salt reductions in the 0−40 cm soil layer. The results showed that deep vertical rotary tillage could reduce the surface soil salt compared with XG−20 treatment, and the salt desalinization rate increased with the increase in tillage depth. Due to the loosened soil structure, tillage can promote the soil vertical infiltration of water and salt. At the same time, the salt reduction caused by the capillary effect was inhibited [13,18,39].

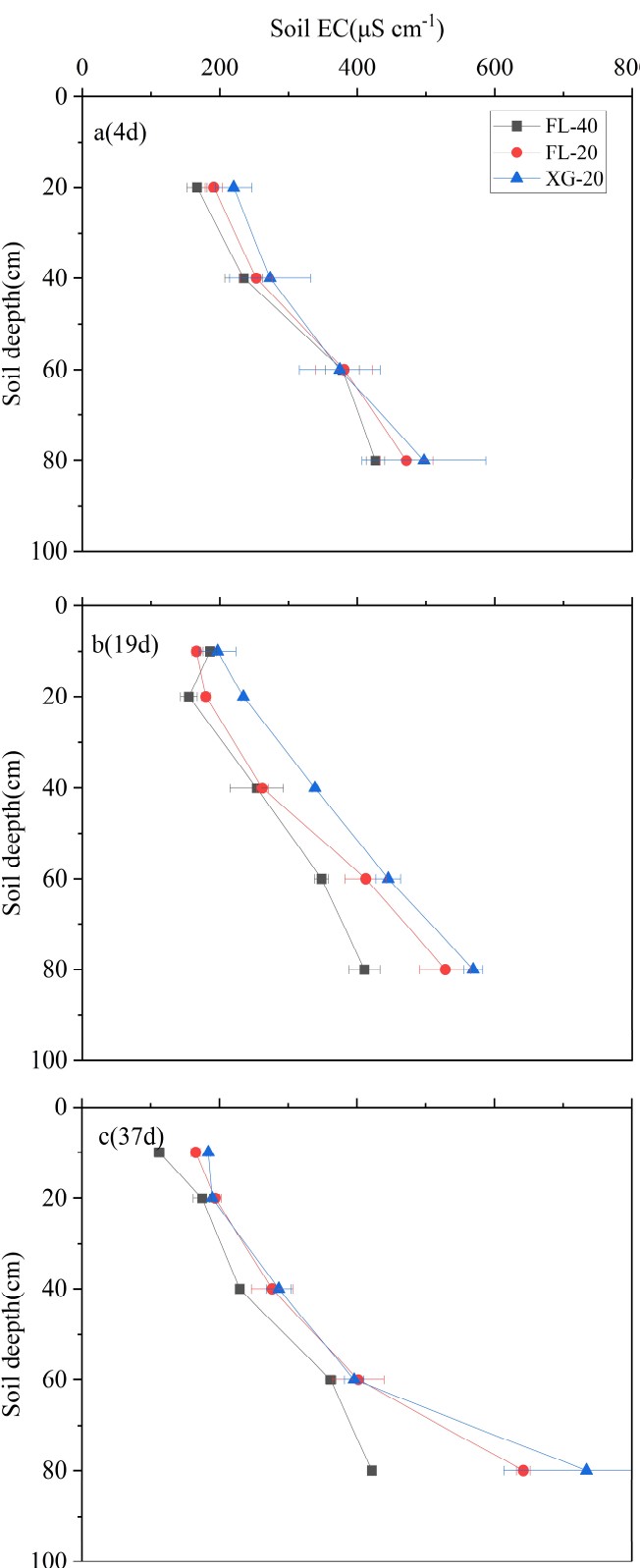

**Figure 6.** Dynamic distributions of EC in the soil profiles under different treatments.

The control of crop root zone salinity is the main purpose of saline soil improvements. This study showed that the deep vertical rotary tillage could reduce the salt content from 0–40 cm in the root zone. Compared with the traditional rotary tillage, the deep vertical rotary tillage treatment can effectively reduce the salt in the surface soil. The deeper the

deep vertical rotary tillage, the more the salt of the corresponding soil layer decreases. This study was consistent with the study of Cao et al. [40]. Water infiltration was hindered in saline soil, and salt was difficult to wash into the underlying soil. Deep vertical rotary tillage promoted soil water infiltration and increased the leaching efficiency of salt [32], which was the main reason for the reducing root zone salt. Other studies obtained the same results [18,40]. In addition, under the action of a strong capillary, salt was easily moved into the surface root zone soil with water. In this paper, it was concluded that deep vertical rotary tillage can reduce the evaporation of soil water and block the rise in capillary water. As a result, the upward movement of salt was reduced, and the desalting phenomenon of surface soil was effectively controlled. Other studies have shown that deep vertical rotary tillage can mix the topsoil into the subsoil, and effectively reduce salt in the topsoil [18]. This was another important reason to control salinity in the root zone. In terms of the cultivation depth of the deep vertical rotary tillage, the deeper the tillage depth, the better the salt reduction effect. This is closely related to the deep and loose tillage layer. Cao et al. concluded, through field experiments, that the salt reduction efficiency of deep vertical rotary was higher than that of traditional rotary tillage, and this was proportional to tillage depth. However, the shallower plough layer of traditional tillage reduced the efficiency of salt leaching because it hinders water infiltration. The soil block was large and unevenly distributed, with more macro-pores, and this preferential flow leads to insufficient salt leaching. The soil conductivity of FL−40 at 10 cm was significantly lower than that in the rotary tillage treatment. At the soil layer of 40 cm, the salinity of FL−20 and FL−40 cm in the 40 cm layer was higher than that in XG−20. The difference in salt content between XG−20 and FL-20 tillage layer may be due to the influence of the shallow water table. A large number of studies have shown that surface salt accumulation is significantly affected by groundwater when the groundwater level is relatively shallow and the salinity is high [41,42]. The deep vertical rotary tillage reduced the evaporation of surface soil water, weakening the capillary effect and reducing the upward movement of salt with water. As a result, soil salinities of 40 and 60 cm were lower than those of traditional rotary tillage. Under shallow water-level conditions, the deep vertical rotary tillage can reduce soil surface evaporation, reduce groundwater level and reduce the salt caused by rising capillary water.

*4.4. Salt Reduction under the Different Tillage Methods*

Reducing root zone salt is an important way of alleviating crop salt stress and improving saline soil [43,44]. In this study, FL−40 treatment had the lowest average salt content in 0−40 cm. The desalinization rate of FL−40 was significantly higher than that of the other two treatments in the 10 cm soil layer (Figure 7). However, the desalinization rate difference between the three treatments did not reach a significant level. The desalinization rate of FL−40 was still the maximum of the three treatments in 40 cm. The desalinization rate of each treatment became negative at 60 cm, indicating that salt began to accumulate in this layer. The difference between the three treatments reached a significant level, and the desalinization rate was FL−40>FL−20>XG−20. In general, the average desalinization rates of the three treatments in 0−60 cm soil layer were 13.15%, 13.31 and 29.47%, respectively. There was little difference in the average desalinization rates of FL−20 and XG−20 treatments. The average desalinization rates of FL−40 treatments were 13.99% and 16.32% higher than those of the first two treatments in the 0−60 cm soil layer. Therefore, it can be concluded that tillage depth was an important factor to improve the desalination rate. It may be that a higher tillage depth improved water infiltration capacity [14]. Thus, compared with 20 cm tillage treatment, FL−40 treatment could more easily promote the leaching of surface salt to deep soil.

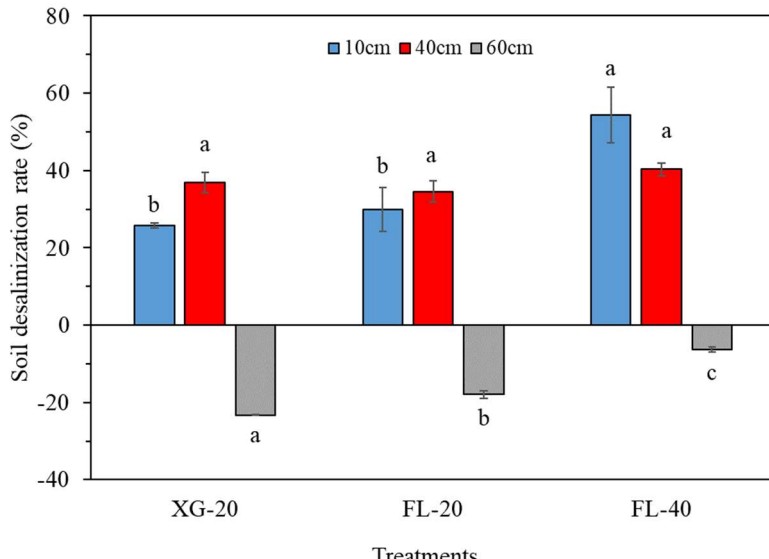

**Figure 7.** Desalinization rates of soils under different treatments.

The desalinization rate is an important index for evaluating the effect of soil salinity leaching. It was found that the desalinization rate of the deep vertical rotary tillage was significantly higher than that of traditional rotary tillage. The desalinization rate of FL-40 treatment in the 10 cm soil layer was significantly higher than that of the other two treatments. It can be seen that the desalinization rate of the surface layer increases with the increase in the depth of deep vertical rotary, which was consistent with the research of Cao et al. The desalinization rate of FL-40 was higher than that of traditional rotary tillage. With the deepening of tillage depth, there is more salt migration downward to plow deep soil layers [21]. As seen in Figure 7, salt accumulation began in the 60 cm soil layer in each treatment, and the deep vertical rotary tillage accumulation degree was lower than that of the other two treatments. The decrease in salt accumulation was caused by a combination of irrigation and reduced evaporation. The results of field experiments showed that after spring irrigation, with the deepening of the deep vertical rotary tillage, the salt pressure effect was better, and the leaching effect of salt in each soil layer was better than the traditional method.

*4.5. Analysis of Different Tillage Methods and Their Application*

Soil tillage is an important measure to regulate soil water, fertilizer, air and heat. Improving soil physical and chemical properties and increasing soil water use efficiency through improving tillage measures is also an important direction in water−saving agriculture research [13]. In this study, the dynamics of soil water and salinity were different between traditional rotary tillage and deep vertical rotary tillage. Deep vertical rotary tillage significantly increased soil water content and reduced soil water evaporation. The studies have shown that deep tillage improves the physical, biological and hydraulic properties of soil [4]. Compared with traditional rotary tillage, the soil water storage depth was further increased with the increase in tillage depth [40]. Traditional rotary tillage could not break through the 20−40 cm soil layer, which hindered rainwater infiltration and the leaching of salinity on the surface, so it is easy to accumulate salinity in the soil layer of 30−50 cm. Therefore, it is necessary to use deep tillage at a 30−50 cm depth to improve soil properties in the context of multi−year conservation agriculture. Deep vertical rotary tillage can cut off the soil capillary and weaken the evaporation of soil water, so as to effectively control soil salinization.

This experiment was carried out indoors without vegetation coverage, and the natural conditions, such as atmosphere and groundwater, were different from those in the field. The temporal−spatial dynamic distributions of soil water and salinity might be different

from those in the field. Therefore, the model requires the further validation of field data. As the coastal saline alkali land is affected by seawater intrusion, the underground water level is shallow, and the salinity is high. Deep vertical rotary tillage, combined with dark pipe drainage measures, may be better able to eliminate the saline barrier. On the other hand, the dark pipe plays a role in controlling the water table and preventing the accumulation of salt in the surface soil caused by the underground water level. This experiment only involved the effects of different tillage methods on soil water and salt movement through the laboratory soil column test. In the future, field experiments will be carried out to verify the experimental results and further study the dynamics of water and salt under different tillage methods and dark pipe combinations.

## 5. Conclusions

The spatial−temporal dynamic distributions of soil water and salt using different tillage methods were studied with the soil column experiment. The results showed that FL−40 effectively improved the soil water infiltration performance and changed soil water distribution. With the evaporation, although the surface soil water content decreased faster than traditional rotary tillage, it could still preserve the water content of the 40−60 cm soil layer. Fl−40 increased soil water content between 40 and 60 cm. Fl−40 effectively promoted the leaching of soil surface salt and increased the desalting rate. At the same time, this reduced soil evaporation and inhibited the accumulation of soil salt. The desalting rate of Fl−40 was significantly higher than other treatments in a 10 cm soil layer, and the salt accumulation was minimal at the 60 cm soil layer. Fl−40 provided a new means of providing the optimal control of water and salt in coastal saline soil.

**Author Contributions:** Conceptualization, J.Y. and W.L.; validation, W.L.; formal analysis, W.L.; investigation, W.L.; resources, R.Y., and X.W.; data curation, W.L.; writing—original draft preparation, W.L.; writing—review and editing, W.L., J.Y., W.X., C.T., X.L.; visualization, W.L.; project administration, W.L., J.Y., W.X. All authors have read and agreed to the published version of the manuscript.

**Funding:** This research was funded by [the National Key Research and Development Programme of China] grant number [2021YFD1900602, 2021YFC3201201] And The research was funded by [the National Natural Science Foundation of China] grant number [41977015].

**Data Availability Statement:** Not applicable.

**Conflicts of Interest:** The authors declare no conflict of interest.

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
