# Peer review of "The Temporal–Spatial Dynamic Distributions of Soil Water and Salt under Deep Vertical Rotary Tillage on Coastal Saline Soil"

_water, doi:10.3390/w14213370_

Round 1
Reviewer 1 Report (New Reviewer)
This study has important implications for water and salinity management in coastal areas. However, there are still some problems:
L17-24: Are these increases or decreases significant? There is no statistical analysis.
L26-28: Is the purpose of this article to see the effect of the model? Or compare different treatments? If it is the latter, the conclusion should highlight which tillage pattern is better. Models are just means.
L107: How many replicates per treatment? How many soil columns in total? From the graphs and tables in result part, there seems to be no replicate.
L199: Statistical analysis should be written with specific statistical methods. Overall, the statistical method is very simple. It is suggested to reanalyze the data and add ANOVA. What exactly are statistical tests in line 200?
L202-318: The discussion is not in-depth enough, and the causes and mechanisms of the changes and differences need to be analyzed or speculated.
L320-325: Just write the conclusion.
Author Response
Please see the attachment.

Reviewer 2 Report (New Reviewer)
Manuscript ID: water-1915823
Manuscript Title: Soil column experiment and simulation for the temporal-spa-tial dynamic distributions of soil water and salt under deep vertical rotary tillage on coastal saline soil
Comments:
This work focused on Soil column experiment and simulation for the temporal-spa-tial dynamic distributions of soil water and salt under deep vertical rotary tillage on coastal saline soil. This article is well written, technically sound and in well scientific style. However, manuscript needs to address certain critical points before recommending it for publication.
1. The title of present study is a little long please try concise it.
2. Please expression of key words looks not good revise it carefully as per your study plan.
3. Novelty point of view, how this research work is unique from already published articles.
4. The discussion section is not well written, please update with few latest references as per your results.
5. The author has not performed literature cited carefully in this manuscript.
6. The conclusion is very poor written please extend a little.
7. This paper must be edited thoroughly for language (vocabulary, grammar and syntax) and scientific notation, preferably by a professional editing service, or a third party who is fluent in English. Their contribution should be recognized in an "Acknowledgements" section.

Round 2
Reviewer 1 Report (New Reviewer)
Compared with the previous MS, the structure of the article, research significance and purpose, and results display have been greatly improved, but there are still some problems in statistics.
L156-157: please add the full name of the ANOVA. What are the specific ANOVA parameters? Has the premise of ANOVA been tested? What multiple comparisons do you use? We need more details on ANOVA. In addition, still no statistical significance can be seen in the results. Is the increase or decrease significant at the statistical level?
Author Response
Please see the attachment.

This manuscript is a resubmission of an earlier submission. The following is a list of the peer review reports and author responses from that submission.
Round 1
Reviewer 1 Report
Review of article NO: MDPI Water- 1818198
The manuscript investigated the soil moisture and salt dynamics under conventional rotary tillage and deep rotary tillage in a soil column experiment using the coastal soil of China. This could have been an interesting study under in-situ condition rather than in a soil column study with cylinder diameter of only 15 cm. I was wondering how the conventional rotary tillage and deep vertical rotary tillage can be done in such a small diameter cylinder filled with soil column, which would deviate much from the real field situations. It was mentioned that a hand electric drill with augur was used. But, there is no comparative evaluation or citation of any previous work pertaining to use of such low power drillers as compared to rotary sub-soiler or chisel plough which is generally used for deep ploughing.
Moreover, the lab experiment with soil columns using hand held deep rotary drill with augur should have minimum three replications besides proper statistical design to corroborate the findings for suitable validation in real field situation. However, replication of experiment is not mentioned in the present form of manuscript.
Moreover, mention of deep vertical rotary tillage to be a new approach and citing old publications viz. Mohanty et al., 2007 besides other as in the reference is contradictory and questionable.
Further, the manuscript lack in clarity in presentation of data and focus on the findings of one set of soil coloum experiment with three different treatments without any replications.
The title of the manuscript should indicate that the study was undertaken in a soil coloumn, which is missing in title.
The irrigation applied was as per the cultivation requirement and the mention of soil moisture before and 24 hrs after irrigation is not presented in the manuscript. In real field situation, the irrigation scheduling for a given crop under saline environment will behave differently and deviate much as mentioned in the manuscript.
In Tbale-2, the other treatments as mentioned in not clear.
There is no mention about the percolated saline water and its safe removal through sub surface drainage or other technologies else it will create a strong saline ground water in the coastal region.
The data acquired for calibration and validation of HYDRUS1-D model is interesting activity and the findings from one set of soil column experiment is very encouraging.
Nonetheless, the manuscript in the present form may be rejected and re submission with replicated soil columns or in situ evaluation can be encouraged.
Reviewer 2 Report
The paper by Li et al. evaluates a new tillage method (deep vertical rotary tillage) on the soil properties, i.e., water and salinity regulation in saline-alkali soil in China. The authors used a soil column experiment and numerical simulation to study the characteristics of the spatial-temporal dynamic distributions of water and salinity under different tillage on coastal saline soil. The strength of the article is the evaluation of the new tillage method on the soil properties. The main limitation is poorly written text. The authors should check the text in terms of editorial. There are many editorial shortcomings, some of which I have highlighted below. Although I have no severe comments on the methods applied, I pointed out some suggestions to improve the paper.
Major comments:
L76-82: this paragraph needs more information. What is the hypothesis of this study? Please explain what ‘different tillage’ means.
L85: there is unclear how the soil samples have been collected. Please describe soil sample collection in more detail.
L85-87: I recommend adding the map with the accurate location of the study site
L122-123: why this amount of irrigation was applied? Please explain
Table 2: there is unclear how the smoothness of pore size distribution and pore connectivity have been determined/calculated
Minor comments:
L52: please add the break between saline and alkali
L56: please remove the full stop after the word “pan”
L93: please exchange ‘;’ on the full stop
Figure 1: Please explain what the XG-20, FL-20, FL-40 mean in the figure caption
Table 1: please explain what the EC means in the table caption
Table 2: Note – please organize the explanation of the abbreviations according to the order in which abbreviations appear in the table